# Low Energy Shock Wave Therapy Inhibits Inflammatory Molecules and Suppresses Prostatic Pain and Hypersensitivity in a Capsaicin Induced Prostatitis Model in Rats

**DOI:** 10.3390/ijms20194777

**Published:** 2019-09-26

**Authors:** Hung-Jen Wang, Pradeep Tyagi, Yu-Ming Chen, Michael B. Chancellor, Yao-Chi Chuang

**Affiliations:** 1Department of Urology, Kaohsiung Chang Gung Memorial Hospital and Chang Gung University College of Medicine, Kaohsiung 83301, Taiwan; hujewang@gmail.com; 2Center for Shockwave Medicine and Tissue Engineering, Kaohsiung Chang Gung Memorial Hospital and Chang Gung University College of Medicine, Kaohsiung 83301, Taiwan; corn88504047@gmail.com; 3Department of Urology, University of Pittsburgh, School of Medicine, Pittsburgh, PA 15231, USA; tyagpradeep@gmail.com; 4Department of Urology, Beaumont Health System, Oakland University William Beaumont School of Medicine, Royal Oak, MI 84073, USA; chancellormb@gmail.com

**Keywords:** prostatitis, shock wave, capsaicin, Caspase-1, COX-2

## Abstract

The effect of low energy shock wave (LESW) therapy on the changes of inflammatory molecules and pain reaction was studied in a capsaicin (10 mM, 0.1 cc) induced prostatitis model in rats. Intraprostatic capsaicin injection induced a pain reaction, including closing of the eyes, hypolocomotion, and tactile allodynia, which effects were ameliorated by LESW treatment. LESW therapy (2Hz, energy flux density of 0.12 mJ/mm^2^) at 200 and 300 shocks significantly decreased capsaicin-induced inflammatory reactions, reflected by a reduction of tissue edema and inflammatory cells, COX-2 and TNF-α stained positive cells, however, the therapeutic effects were not observed at 100 shocks treated group. Capsaicin-induced IL-1β, COX-2, IL-6, caspase-1, and NGF upregulation on day 3 and 7, while NALP1 and TNF-α upregulation was observed on day 7. LESW significantly suppressed the expression of IL-1β, COX-2, caspase-1, NGF on day 3 and IL-1β, TNF-α, COX-2, NALP1, caspase-1, NGF expression on day 7 in a dose-dependent fashion. LESW has no significant effect on IL-6 expression. Intraprostatic capsaicin injection activates inflammatory molecules and induces prostatic pain and hypersensitivity, which effects were suppressed by LESW. These findings might be the potential mechanisms of LESW therapy for nonbacterial prostatitis in humans.

## 1. Introduction

Chronic prostatitis and chronic pelvic pain syndrome (CP/CPPS) is a common type of male urological condition, affecting 5–10% of male population. It is a frustrating disease defined as genital urinary pain with or without voiding symptoms in the absence of uropthogenic bacteria or malignancy [1]. The etiology of CP/CPPS is still largely unknown. The possible causes of CP/CPPS may be neurogenic or immunogenic inflammation, which activates prostate afferent nerves and induces inflammation, prostate pain, and referred pain. However, traditional inflammation markers, such as white blood cells in the prostate tissue or prostatic fluid, do not correlate with the severity of the pain [1]. Dysregulated expression of cytokines including INF-g, IL-6, IL-10, TNF-a, and IL-1b in CP/CPPS implicate the role of an autoimmune process in the pathophysiology of CPPS [2]. Limited clinical efficacy of the available treatments for nonbacterial prostatitis [3,4,5] motivates the research for a novel therapy for patients with refractory chronic prostatitis and prostatic pain.

High-energy extracorporeal shock wave (SW) therapy has been used for the treatment of urolithiasis since 1982 [6]. Recent evidence indicates that SW at lower energy enhances the expression of VEGF, eNOS, PCNA, and recruitment of progenitor cells, and enhances tissue regeneration [7]. Furthermore, several studies demonstrate that low energy extracorporeal shock wave therapy significantly improves the pain, quality of life, and voiding conditions in CP/CPPS patients compared with the placebo group [8,9,10,11]. However, the mechanism of low energy shock wave (LESW) on CP/CPPS largely remains unknown. We hypothesize that LESW might suppress the expression of inflammatory molecules and reduce the inflammatory condition and prostatic pain. We evaluated the effects and mechanisms of LESW therapy in a capsaicin-induced prostatitis model in rats.

## 2. Results

### 2.1. Pain Behavior Induced by Intraprostatic Capsaicin Injection Were Suppressed by LESW Treatment

Intraprostatic capsaicin injection induced marked behavior changes in rats, including closing of the eyes and hypolocomotion. These behaviors represent the prostate pain, which was significantly decreased by LESW treatment (Figure 1A). Increased eye open score in capsaicin group (3.25 ± 0.34 vs. vehicle 1.37 ± 0.08, *p* = 0.0001) was significantly decreased after 100, 200, and 300 shockwave treatment (1.92 ± 0.37; 1.46 ± 0.14; 1.37 ± 0.12, respectively, *p* < 0.05 *n* = 8 each group). Locomotion scores in all groups were lower after LESWT compared with the capsaicin group, but not significant in the 100 shock wave group (3.46 ± 0.39; 3.00 ± 0.13; 2.13 ± 0.18; 1.75 ± 0.20, *p* < 0.05, *n* = 8 each group, Figure 1A–C). General characteristics of the experimental animals was provided in Appendix A.

### 2.2. Capsaicin Decrease Pain Threshold and Induce Skin Hypersensitivity, Which Were Suppressed by LESW Treatment

After injection with capsaicin, the rats became more responsive to stimulation and the pain threshold decreased compared to the sham control (30.70 ± 2.43 vs. 63.33 ± 4.90 and 31.25 ± 4.64 vs. 68.22 ± 5.77 g/s, 2-h and 24-h time point, respectively) (Table 1, Figure 1B). Hypersensitivity to pain at the scrotal base was ameliorated by LESW treatment for at least 24 h. The effect of pain relief by LESW was observed in a dose-dependent fashion. The 200 and 300 shockwave groups showed significant pain threshold increment (*p* < 0.0001 and *p* < 0.001, 24-h time point), while 100 shockwaves did not have a significant effect for pain reliving (Table 1, Figure 1B).

### 2.3. LESW Treatment Suppressed Capsaicin-Induced Inflammatory Reaction, and Increased COX-2 and TNF-α Immunostaining

Capsaicin-induced inflammatory cells infiltration and immunoreactivity for COX-2, TNF-α in prostate was significantly higher compared to vehicle injection (Figure 2). The capsaicin-induced prostatic inflammation was dose-dependently ameliorated by LESWT. LESW decreased COX-2(+) cell accumulation on day 3 in 100, 200, and 300 shockwave groups (4.5%, 63.5%, and 54.9% reductions, respectively) (Figure 2L). The decrement was significant in the 200, 300 shock wave groups but not in the 100 shock wave group (*p* = 0.002, 0.0046, and 0.757). The same result was observed on day 7 (Figure 3L). The TNF-α (+) cell count increased with time (9.60 ± 1.92 for day 3 and 26.78 ± 8.36 for day 7, Figure 2R, and Figure 3R). LESW significantly decreased the TNF-α (+) cells only at day 3 in the 300 shock group and day 7 in both the 200 and 300 shock groups (*p* = 0.028).

### 2.4. LESW Suppressed Capsaicin-Induced Upregulation of Inflammatory Molecules

Western blot demonstrated that capsaicin injection induced a significant increase in the expression of inflammatory molecules including COX2, IL-1β, TNF-α, IL-6, NALP1, Caspase1, and neurotrophic factor NGF on day 7 relative to the sham group. LESW treatment significantly decreased the COX2, IL-1β, caspase1, and NGF protein expression on day 3 and day 7. The effect was dominant on the 200 and 300 shock groups. One hundred shockwaves only decreased the COX-2 and caspase1 expression. Capsaicin-induced TNF-α and NALP1 expression was indifferent from control on day 3. LESW did not produce any effect on these molecules. Significant increase in expression of TNF-α and NALP1 on day 7 was ameliorated by LESW. There was no statistically significant difference in the IL-6 level among capsaicin injected group and shockwave groups on day 3 and day 7 (Figure 4 and Figure 5 and Table 2, full original images of western blots in Appendix A).

## 3. Discussion

The current study revealed that intraprostatic capsaicin injection induced pain reaction, tissue inflammation, and hypersensitivity associated with the activation of NALP1, IL-1β, TNF-α, NGF, and COX2 expression at the different time points, which were suppressed by LESW treatment in a dose-dependent fashion.

The pathogenesis of CP/CPPS is incompletely understood. Several pathomechanisms have been proposed, including cryptic infection, autoimmunity, endocrine imbalance, pelvic floor hyperactivity, peripheral and central sensitization, and neuroplasticity, which can contribute to prostatic inflammation and pain [12,13,14]. Current available therapeutic options are far from satisfactory for patients. Therefore, it is urgent to develop a novel and effective therapeutic approach.

Zimmermann et al. firstly reported the feasibility of LESW therapy on CPPS [10]. In addition, they demonstrated that LESWT significantly improved the pain and micturition related to CPPS, as well as erectile function and quality of life at three months in a randomized, double-blind, placebo-controlled trial [11]. Al Edwan et al. further demonstrated the maximal effects of LESW (ESWT once a week for one month) on the IPSS, NIH-chronic prostatitis symptom index (NIH-CPSI), and the American urological association quality of life due to urinary symptoms scale (AUA QoL US) at two weeks, and the therapeutic effects were maintained until 12 months [15]. However, the molecular and histological changes after LESW remained unanswered.

LESW therapy has been applied in a wide range of disorders in orthopedics (bone healing, tendinopathies, and cartilage repair), dermatology (diabetic foot, ulcers, wound healing, and fire burns), and neurology (spasticity or spastic hypertonia, and neuron protection) [16,17,18,19,20,21]. The mechanisms of LESWT include four reaction phases: physical phase, physical–chemical phase, chemical phase, and biological phase. The biological effects of LESW included angiogenesis, tissue regeneration, and anti-inflammation through the modulation of various mechanisms [22].

Anti-inflammatory action of LESW is related to the mechanism of “mechanotherapy” and involves different biological pathways in immunomodulation [23]. Previous reports have demonstrated that LESW significantly decreased the infiltration of inflammatory cells (neutrophils and macrophages) in burned skin [24] or skin flaps [25], and in the cyclophosphamide-induced cystitis [26,27].

The COX-2 enzyme is involved in the synthesis of prostaglandins, which are important mediators of pain and inflammation [28]. After capsaicin injection into the prostate, COX-2 positive inflammatory cells accumulated and increased COX-2 expression [29,30]. The upregulation of COX-2 in prostate can lead to production of prostaglandins, which might trigger capsaicin-sensitive afferent fiber and induce neurogenic prostatitis and prostate pain. The current findings demonstrated that LESW can suppress prostate inflammation through downregulation of COX-2 expression in both cell and protein levels (Figure 2, Figure 3 and Figure 4).

NGF is the key molecule involved in neural regulation, inflammation, and pain. NGF can bind to receptors expressed on mast cell membranes to cause degranulation and cytokine and chemokine release [16]. Previous studies reported that NGF levels in expressed prostate secretions are in proportion to pain severity in CP/CPPS [31]. Capsaicin mediated upregulation of NGF expression in the prostate on day 3 and day 7 was dose-dependently ameliorated by LESW treatment. The effect of LESW on downregulation of COX-2 and NGF expression after capsaicin injection was in correspondence with decrement of pain behavior and increment of pain threshold.

Though COX-2, IL-1β, and IL-6 are regarded as inflammatory index, and might be related to prostate pain and inflammation, they are not in the same regulatory mechanisms and might not be synchronously regulated by the stimulation. We found LESW had a limited effect on IL-6 expression in this study. Our previous study revealed that the protein level of IL-6 in cyclophosphamide induced cystitis in rats was significantly decreased at day 4 but had no significant change at day 8 after LESW treatment [27]. IL-6 is produced at the site of inflammation and plays a key role in the acute phase response. When monocytes and macrophages are stimulated, a range of signaling pathways including NF-κB or MAPKs enhance the transcription of the mRNA of inflammatory cytokines, such as IL-6, TNF-α, and IL-1β. TNF-α and IL-1β also activate transcription factors to produce IL-6 [32]. Jeon et al. recently showed LESW therapy decreases COX-2 by inhibiting TLR4-NFκB pathway in a prostatitis rat model. TRAF2 regulator in ERK1/2 inhibition significantly reduced inflammation [33]. The above-mentioned signaling events co-operatively induce inflammation with different level of the expression of IL-1β, IL-6, and other inflammatory molecular through different stimulation models.

Several cytokines, including IL-1β, TNF-α, IL-6, IL-8, IL-10, and inflammasome NALP1, and caspase 1, have been found to be important in the pathogenesis of CP/CPPS [13,34,35]. LESW treatment has been shown to suppress the proinflammatory cytokines (IL-1β, IL-4, IL-6, IL-10, IL-12, IL-13, and INF-r), matrix metallopeptidase (MMP-3, MMP-9, MMP-13) and chemokines (CCL2, CCL3, CCL4, CCL7, CXCL1, CXCL2, CXCL5) in murine skin isografts, burned skin, and urinary bladder [24,26,36]. Capsaicin mediated upregulation of IL-1β and caspase 1 was noted on day 3 and day 7 following injection, while upregulation of TNF-α and NALP1 was noted only on day 7.

NALP1 is responsible for inflammasome assembly and caspase-1 activation for cleavage of IL-1β and IL-18 precursors into mature forms [37,38]. Chen et al. found that an increase in NALP1 and caspase 1 expression is responsible for persistent inflammation in the carrageenan-induced prostatitis model in rats. The current study demonstrated that LESW is able to attenuate NALP1 and caspase 1 expression and its downstream cytokine IL-1β, which suggests that LESW has an impact on the inflammasome pathway [13]. It has been suggested that the IL-1β and TNF-α level were increased in expressed prostatic secretions from men with CPPS [35]. TNF-α and IL-1β can further activate transcription factors to produce COX-2 and IL-6 through TLR4-NFkB and MAPKs pathway. Physical disruption of NALP-1 inflammasome by LESW has a multi-pronged action on inflammation signaling (Figure 6).

The limitation of the current study includes the duration of capsaicin injection evoked pain and inflammation is less than 1 week, which cannot completely reflect the chronic character of CP/CPPS in men. Currently, several rodent models have been developed, each with its own unique characteristics and limitations [39]. Capsaicin intra-prostatic injection induced neurogenic prostatitis and acute pain, which might reflect the acute stage or flare-up of neurogenic prostatitis, which are important elements of nonbacterial prostatitis [30].

## 4. Materials and Methods

### 4.1. Animals

All experimental procedures were performed in male Sprague–Dawley (SD) rats weighing 300 to 350 gm. Animals were housed under constant temperature and humidity, and under a 12-h light and dark cycle. All the following procedures were reviewed and approved by the institutional animal care and use committee of Chang Gung memorial hospital (IACUC no. 2016012701, approval date: 20170226) and complied with the NIH guide for care and use of laboratory animals. At the end of experiments, animals were deeply anaesthetized by intramuscular injection of Zoletil 50 (25~50 mg/kg) and Xylazine (10~23 mg/kg) for humane sacrifice followed by transcardiac perfusion with Krebs buffer.

### 4.2. Capsaicin-Induced Prostatitis

0.1 mL of vehicle of 10% alcohol, 10% Tween 80, and 80% saline with or without Capsaicin 10 mM (sham control group) was injected into each of the ventral lobe of prostate with a 30-gauge needle [29].

### 4.3. Low Energy Shock Wave Treatment

Animals were assigned into five groups (*N* = 8 for each group). The sham control group received low laparotomy and intraprostatic vehicle injection without any shockwave therapy. The other four groups had laparotomy and intraprostatic capsaicin injection. In these four prostatitis groups, one did not receive any shock wave therapy, and the other three groups had transcutaneous LESW therapy in different energy flux density (100, 200, or 300 pulses; 0.12 mJ/mm^2^; frequency of two pulses per second) immediately and 24 h after capsaicin injection.

Under 2% isoflurane anesthesia, low laparotomy was performed to explore the prostate for capsaicin injection. The skin was closed in layers, and the shock wave probe (SD-1, Storz, Tägerwilen, Germany) was gently placed over the skin surface above the prostate area after application of ultrasound transmission gel.

### 4.4. Assessment of Pain Behaviors

The rats were brought into the laboratory the day before prostate injection and placed in individual observation boxes. Baseline behavior and behavioral changes following second shock-wave therapy was scored every 10 min for 3 times after recovery from anesthesia. As described previously [27], we used a scoring scale of 1 to 5. A minimum score of 1 was assigned if no parameter was affected, compared with baseline behavior and maximum symptom severity, which was assigned the score of 5. For eye movement, the scores assigned were 1 for normal opening, 5 for complete closing, 3 for half-closed eyes, 2 and 4 for the 2 intermediate positions between open and half-closed, and between half-closed and closed, respectively. For locomotion, a score of 5 was given for complete limpness of the hind limbs or motionless status during the observation period. A score of 1 was assigned when locomotion did not significantly change from the pre-injection condition. Blinded observers performed all the experiments and each experimenter scored two rats in parallel.

### 4.5. Von Frey Filament in Behavioral Testing

The rats were tested for tactile allodynia at 2 h and 24 h after pain behavior assessment [13]. Referred hyperalgesia and tactile allodynia were quantified using von Frey filaments applied to the scrotal base. Electronic von Frey aesthesiometer (Ugo Basile Srl, Varese, Italy) with stiff large size filaments was used. Stimulation was performed when the rat remained quiet with the scrotum resting on the bottom of cage. An average of three stimulations every 10 min was used for analysis.

### 4.6. Histology and Immunohistochemistry

Prostate harvested 3 and 7 days after capsaicin injection was divided into two parts; one part was fixed in buffered formaldehyde 10% for 24–48 h, and then embedded in paraffin, for hematoxyline and eosin staining and immunohistochemistry of inflammasome using commercially available kit (Thermo Scientific UltraVision™ Quanto Detection system, Fremont, CA, USA). The other part was frozen in liquid nitrogen and preserved for Western blot. Inflammatory cells were quantified as density per unit area by random counting on 4 spots under a high-power field (200 magnification). Prostate tissue sections for immunohistochemistry were cut into 3 μm sections, de-waxed in xylene, and rehydrated with tap water through decreasing concentrations of alcohols. Antigen retrieval was achieved by pressure cooking tissue sections immersed in 10 mM citrate buffer (pH 6.0). Endogenous peroxidase activity was blocked with 3% hydrogen peroxide solution. Tissue sections were then incubated with primary polyclonal rabbit against COX-2 antibody (Cayman Chemical, Ann Arbor, Michigan, MI, USA), TNF-a (SAB4502982 Sigma–Aldrich, St. Louis, MO, USA), at dilutions of 1:1000, 1:200, respectively, then diluted in antibody diluent solution (Zymed^®^, Thermo Scientific, Carlsbad, CA, USA) for 30 min at room temperature. Sections were washed in PBS (pH 7.0) after incubation in primary antibody and then incubated in BioGenex^®^ Super Enhancer^TM^ Reagent for 20 min. After further washing in PBS, sections were incubated for 20 min in Bio- Genex Polymer Horseradish Peroxidase Complex, followed by a PBS wash. Slides were developed with 3,3-diaminoben-zidine chromogen (BioGenex DAB substrate) and counter-stained with Mayer’s hematoxylin. Slides were then dehydrated through increasing concentrations of alcohol to xylene and coverslip mounted with Entellan^®^.

### 4.7. Western Blotting Analysis

Prostate protein was extracted from grounded frozen tissue for Western blot analysis of proteins expression according to the standard protocol (Amersham Biosciences^TM^, Buckinghamshire, UK). The samples were homogenized in protein extraction solution (T-PER; Thermo Scientific, Rockford, IL, USA) prior to sonication and purification. The amount of total protein was measured with the Bradford protein assay method (Bio-Rad Laboratories, Hercules, CA, USA). SDS-polyacrylamide gel electrophoresis (PAGE) was performed using the buffer system of Laemmli. Briefly, an aliquot of the extracts equivalent to 50 µg protein was loaded onto 4–20% Mini-PROTEAN^®^ TGX™ Precast Gels (Bio-Rad Laboratories, Hercules, CA, USA), electrophoresed at a constant voltage of 100V for 1.5 h and transferred to Hybond-P PVDF Membrane (Amersham Biosciences). The membrane was blocked with 5% skim milk powder (Fluka, Sigma–Aldrich, St. Louis, MO, USA) for 60 min and then immunoblotted overnight at 4 °C with mouse anti-actin monoclonal antibody (Chemicon, 1:10,000 dilution), primary antibody against COX-2 (ab15191, Abcam, Cambridge, MA, USA), TNF-α (ab6671, Abcam, Cambridge, MA, USA), IL-1β, (ab9722, Abcam, Cambridge, MA, USA), IL-6 (#ARC0062, Thermo Scientific, Rockford, IL, USA), Caspase 1, (ab1872, Abcam, Cambridge, MA, USA), NGF (ab52918, Abcam, Cambridge, MA, USA), and NALP1 (ab3683, Abcam, Cambridge, MA, USA), respectively. After being washed, the membrane was incubated with secondary antibody using 2.5% defatted milk powder in TBS for one hour at room temperature using a horseradish peroxidase-linked anti-rabbit or anti-mouse, anti-rat immuoglobulin G (diluted concentration 1:5000–1:10,000). Western blots were visualized by enhanced chemiluminescence (ECL) detection system (Amersham Biosciences). The amount of β-actin AC-15 HRP (1:40,000 dilution; ab49900, Abcam, Cambridge, MA, USA) was also detected as the internal control. Quantitative analysis was done using LabWorksTM Image Acquisition and Analysis software Image Lab (Bio-Rad Laboratories, Inc., Berkeley, CA, USA).

### 4.8. Statistical Analysis

Quantitative data are expressed as mean plus or minus standard error of mean. Statistical analyses were performed using two-way ANOVA, with Bonferroni post-test where applicable, with *p* < 0.05 considered significant. Statistical analyses were performed using the GraphPad Prism 7. Software, Inc. for biostatistics. (GraphPad Prism, La Jolla, CA, USA).

## 5. Conclusions

In conclusion, LESW treatment inhibited NALP1, caspase1, IL-1β, TNF-α, COX-2, and NGF expression and reduced the pain and inflammation in capsaicin-induced prostatitis in rats. This evidence of LESW mediated behavioral and molecular changes support the promise of LESW as a treatment for CP/CPPS in humans.

## Figures and Tables

**Figure 1 ijms-20-04777-f001:**
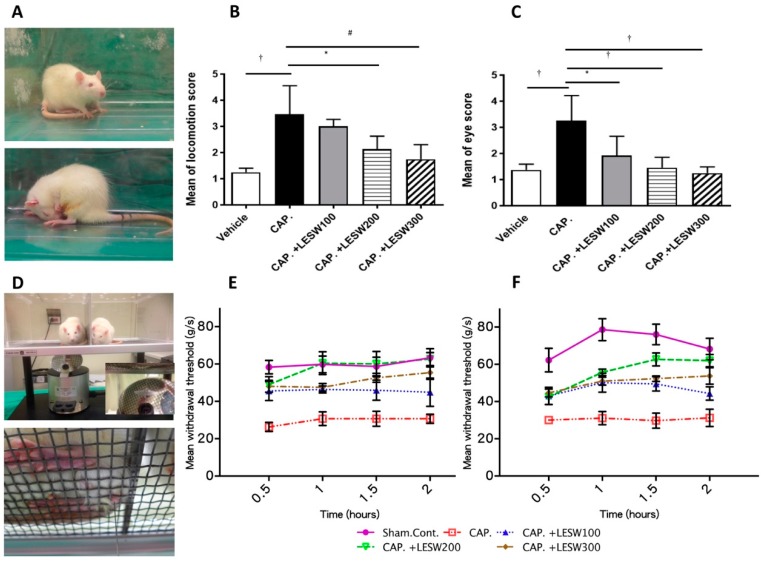
Responses of pain behaviors and sensitivity to intra-prostatic capsaicin and LESW therapy. (**A**): Vehicle injected animal (upper panel) and prostatic pain induced by capsaicin injection, characterized by closing of the eyes and hypo-locomotion (lower figure); (**B**): eye open score time course after second LESW treatment; (**C**): locomotion score time course after second LESW treatment. (**D**): Mechanical allodynia confirmed by Von Frey filament examination. Von Frey filaments with different bending forces were delivered to scrotum. (**E**,**F**): Mean withdrawal threshold force to Von Frey stimulation assessed 2 h (**E**) and 24 h (**F**) after second LESW therapy. Capsaicin-induced reduction of pain threshold and hypersensitivity, which effects were decreased by LESW treatment in a dose-dependent fashion. * *p* < 0.05; # *p* < 0.01; † *p* < 0.001.

**Figure 2 ijms-20-04777-f002:**
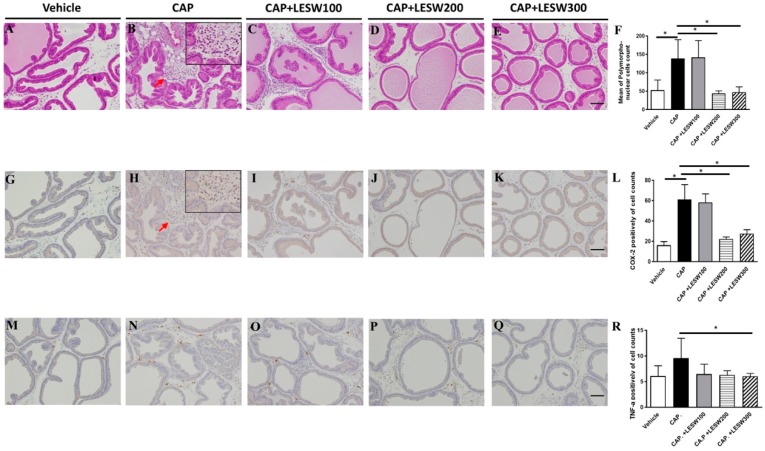
Histology and immunohistochemistry staining of prostate on day 3 post-capsaicin injection with or without LESW treatment. (**A**): sham control, (**B**): capsaicin (CAP) injected into prostate glandular showed edematous change with accumulation of inflammatory cells (red arrow) in the interstitial space, which effects were decreased in the 200 and 300 shocks LESW treatment groups ((**F**), average cell count per unit area). COX-2 and TNF-α positive inflammatory cells (red arrow) accumulated in CAP injected group (**H**,**N**). Compare with capsaicin prostatitis (CAP) group, 200 and 300 shocks of LESW significantly decreased COX-2 positive cells (**L**). Only 300 shocks of LESW significantly decreased TNF-α positive cells compared with CAP group (**R**). (**A**–**E**): hematoxyline and eosin stain; (**G**–**K**): COX-2 stain; (**M**–**Q**): TNF-α stain; magnification ×100; inlet, magnification ×400; scale bars indicate 100 μm. * Statistically significant difference between groups (*p* < 0.05) (**F**,**L**,**R**).

**Figure 3 ijms-20-04777-f003:**
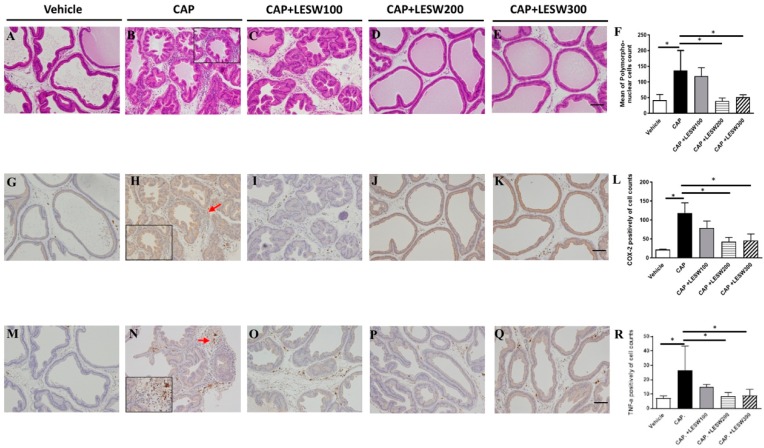
Histology and immunohistochemistry staining of prostate on day 7 post-capsaicin injection with or without LESW treatment. (**A**): sham control, (**B**): day 7 post-capsaicin (CAP) injection showed edematous change with accumulation of inflammatory cells in the interstitial space, the effects of which were decreased by 200 and 300 shocks LESW treatment (**D**,**E**), while 100 shocks therapy (**C**) showed no significant effect ((**F**), average cell count per unit area). COX-2 and TNF-α positive inflammatory cells accumulated (red arrow) in CAP group (**H**,**N**) was significantly decreased by LESW 200 and 300 shocks treatment (**L**,**R**) (*p* < 0.05, CAP vs. 200 SW group; CAP vs. 300 SW group). (**A**–**E**): hematoxyline and eosin stain; (**G**–**K**): COX-2 stain; (**M**–**Q**): TNF-α stain; magnification ×100; inlet, magnification ×400; scale bars indicate 100 μm. * Statistically significant difference between groups (*p* < 0.05) (**F**,**L**,**R**).

**Figure 4 ijms-20-04777-f004:**
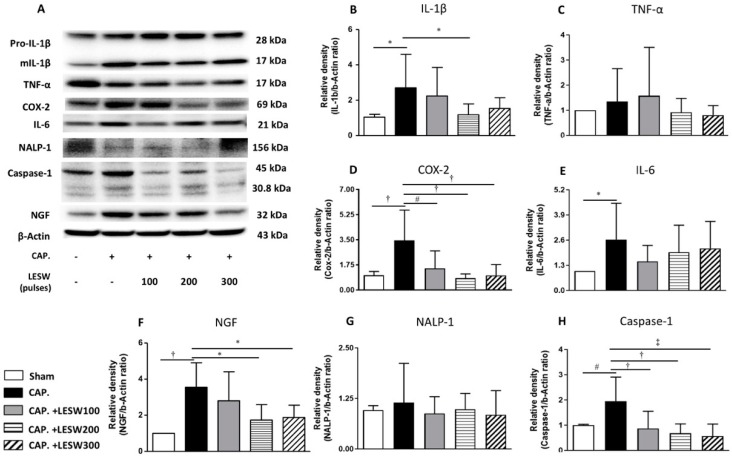
Day 3. Western blot for detecting IL-1β, TNF-α, COX-2, IL-6, inflammasome NALP-1, caspase-1, and NGF expression in the prostate (**A**). Capsaicin-induced upregulation of IL-1β, COX-2, IL-6, caspase-1, and NGF expression (**B**,**D**–**F**,**H**). LESW significantly suppressed IL-1β, COX-2, caspase-1, NGF expression in a dose-dependent manner but not in IL-6, TNF-α and NALP-1 (**C**,**E**,**G**) (100 shocks: COX-2, caspase-1; 200 shocks: IL-1β, COX-2, caspase-1, NGF; 300 shocks: COX-2, caspase-1, NGF) *, *p* < 0.05; # *p* < 0.01; † *p* < 0.001; ‡ *p* < 0.0001.

**Figure 5 ijms-20-04777-f005:**
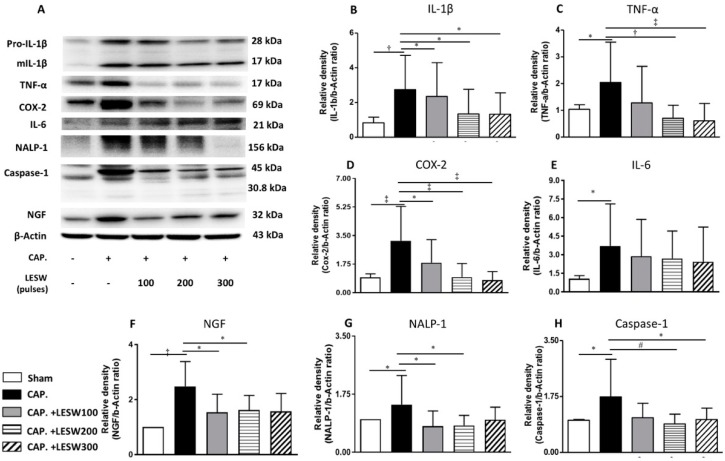
Day 7. Western blot for detecting IL-1β, TNF-α, COX-2, IL-6, inflammasome NALP-1, caspase-1, and NGF expression in the prostate(**A**). Capsaicin-induced all inflammatory molecules upregulation (**B**–**H**). LESW significantly suppressed IL-1β, TNF-α, COX-2, NALP1, caspase-1, NGF expression in a dose-dependent manner. The protein amount of IL-6 was insignificantly decreased in the capsaicin treated prostate. * *p* < 0.05; # *p* < 0.01; † *p* < 0.001; ‡ *p* < 0.0001.

**Figure 6 ijms-20-04777-f006:**
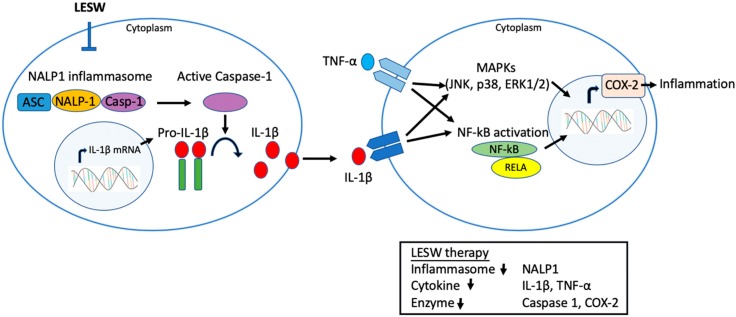
A model illustrating LESW treatment inhibiting the inflammatory molecular in capsaicin induced prostatitis in rats. LESW application disrupted the assembly of multimolecular complex of NALP-1 inflammasome, which leads to decrease caspase 1 activation and the production of the pro-inflammatory cytokines IL-1β. The downregulated IL-1β and TNF-α bind to their cognate receptors to activate downstream signaling, including various mitogen-activated protein kinases (MAPKs) and cytoplasmic nuclear factor-kappa B (NF-κB), by which COX-2 expression would be decreased in the prostate. Physical disruption of NALP-1 inflammasome by LESW has a multi-pronged action on inflammation signaling.

**Table 1 ijms-20-04777-t001:** Mechanical allodynia evaluated by von Frey filament examination (*n* = 8 for each group) *, *p* < 0.05; **, *p* < 0.01; ***, *p* < 0.001; **** *p* < 0.0001.

			Mean Withdrawal Threshold (g/s) Mean ± SEM				Significant (Tukey’s Multiple Comparisons Test)			
	Time (hour)	Sham	Capsaicin	Cap + LESW 100	Cap + LESW 200	Cap + LESW 300	Cap vs. Sham	Cap + LESW 100 vs. Cap	Cap + LESW 200 vs. Cap	Cap + LESW 300 vs. Cap
**2 h after 2nd LESW**	0.5	58.34 ± 3.60	26.32 ± 2.41	45.56 ± 5.13	49.03 ± 4.16	48.07 ± 3.18	****	*	**	**
	1	59.78 ± 4.43	30.70 ± 3.67	46.39 ± 1.88	60.42 ± 6.11	47.50 ± 2.03	****	*ns*	****	*
	1.5	58.69 ± 4.95	30.70 ± 4.03	45.97 ± 5.34	60.00 ± 6.68	52.60 ± 2.39	***	*ns*	****	**
	2	63.33 ± 4.90	30.70 ± 2.43	44.86 ± 7.58	62.64 ± 3.49	55.42 ± 3.65	****	*ns*	****	***
**24 h after 2nd LESW**	0.5	62.22 ± 6.34	30.00 ± 1.64	42.99 ± 4.64	42.57 ± 4.11	44.64 ± 2.15	****	*ns*	*ns*	*ns*
	1	78.61 ± 5.88	31.11 ± 3.53	50.14 ± 5.09	55.69 ± 1.73	50.94 ± 2.09	****	*	***	**
	1.5	76.04 ± 5.54	29.72 ± 4.03	49.45 ± 3.83	62.64 ± 3.49	52.29 ± 1.47	****	**	****	***
	2	68.22 ± 5.77	31.25 ± 4.64	44.31 ± 3.56	62.00 ± 3.30	53.75 ± 4.40	****	*ns*	****	***

**Table 2 ijms-20-04777-t002:** Western blotting expression of inflammatory molecular in capsaicin and LESW group relative to sham control group on days 3 and 7 (*n* = 8 for each group).

							Adjusted *p*-Value (Bonferroni’s Multiple Comparisons Test)			
		Sham	Capsaicin	Cap + LESW 100	Cap + LESW 200	Cap + LESW 300	Cap vs. Sham	Cap + LESW 100 vs. Cap	Cap + LESW 200 vs. Cap	Cap + LESW 300 vs. Cap
**Day 3**	IL-1β	1.00 ± 0.00	2.7 ± 1.89	2.24 ±1.62	1.18 ± 0.61	1.53 ± 0.61	0.0196	>0.9999	0.0418	0.2519
	TNF-α	1.00 ± 0.00	1.34 ± 1.31	1.58 ± 1.92	0.92 ± 0.55	0.81 ± 0.38	>0.9999	>0.9999	>0.9999	>0.9999
	COX-2	1.00 ± 0.00	3.45 ± 2.11	1.50 ± 1.22	0.82 ± 0.30	1.01 ± 0.76	<0.0001	0.0016	<0.0001	<0.0001
	IL-6	1.00 ± 0.00	2.63 ± 1.86	1.50 ± 0.83	1.98 ± 1.40	2.17 ± 1.40	0.0286	0.3392	>0.9999	>0.9999
	NGF	1.00 ± 0.00	3.55 ± 1.36	2.80 ± 1.61	1.73 ± 0.86	1.88 ± 0.68	0.0004	>0.9999	0.0331	0.0465
	NALP1	1.00 ± 0.00	1.15 ± 0.97	0.87 ± 0.42	0.98 ± 0.39	0.85 ± 0.60	>0.9999	>0.9999	>0.9999	>0.9999
	Caspase-1	1.00 ± 0.00	1.95 ± 0.95	0.88 ± 0.68	0.68 ± 0.37	0.57 ± 0.48	0.0041	0.0008	0.0001	<0.0001
**Day 7**	IL-1β	1.00 ± 0.00	2.75 ± 1.97	2.35 ± 1.95	1.34 ± 1.41	1.33 ± 1.23	0.0003	>0.9999	0.0415	0.0332
	TNF-α	1.00 ± 0.00	2.04 ± 1.51	1.28 ± 1.37	0.71 ± 0.48	0.61 ± 0.65	0.0104	0.2992	0.0002	<0.0001
	COX-2	1.00 ± 0.00	3.18 ± 2.11	1.83 ± 1.41	0.95 ± 0.84	0.77 ± 0.52	<0.0001	0.0173	<0.0001	<0.0001
	IL-6	1.00 ± 0.00	3.71 ± 3.40	2.88 ± 2.97	2.69 ± 2.23	2.43 ± 2.79	0.0265	>0.9999	>0.9999	>0.9999
	NGF	1.00 ± 0.00	2.48 ± 0.90	1.53 ± 0.66	1.62 ± 0.53	1.57 ± 0.65	<0.0001	0.0154	0.0354	0.0507
	NALP1	1.00 ± 0.00	1.43 ± 0.88	0.78 ± 0.46	0.81 ± 0.30	0.98 ± 0.38	0.0255	0.0233	0.0498	0.2945
	Caspase-1	1.00 ± 0.00	1.74 ± 1.18	1.09 ± 0.45	0.89 ± 0.31	1.03 ± 0.36	0.0211	0.0581	0.0055	0.0276

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
