# Peer review of "Low Energy Shock Wave Therapy Inhibits Inflammatory Molecules and Suppresses Prostatic Pain and Hypersensitivity in a Capsaicin Induced Prostatitis Model in Rats"

_ijms, 2019, doi:10.3390/ijms20194777_

Round 1

Reviewer 1 Report

This article is about the effect of low energy shock wave on the prostatitis. In this manuscript, authors established a prostatitis rat model by capsaicin induction and explored the mechanisms of shock wave for nonbacterial prostatitis. This research is interesting and meaningful for the prostatitis study. But this manuscript was short of credible results and was self-contradictory in discussion. Here are some questions authors need to address:

In the “animal”, why authors need to use two anesthetics? And at 4.3 a third anesthetic appeared. Please confirm it. Please clear the group in detail in the method. In 4.3, authors said under anesthesia rat was accepted the low laparotomy for assessing the prostate. Were all rats got this administration? The pain of operation could disturb the assessment of pain behavior and reduce the accuracy. In Result, authors should show the mean weight of each group before and after treatment. In figure 1, figure 1 E and F were not match with captions. Please revise. In figure 2 and 3, the statistically significant difference between which groups should be stated clearly. And in this part authors mentioned cell count but in the method I can find any method or software for counting cells. In figure 4 and 5, IL-1β and IL-6 showed a contrary result after SW treatment. But in discussion authors said LESW suppressed them in earlier studies. Please discuss it in manuscript. Please modify figure 6 more visualized, it’s hard to understand especially with the confusing caption. Grammar and words need to be improved, like page 2, line 61, 62 and 69.

Reviewer 2 Report

To my knowledge, there are so many articles about LESW therapy, but few of them explored the mechanisms. In the manuscript, the authors focus on the effect of low energy shock wave (LESW) therapy on the changes of inflammatory molecules and pain reaction using capsaicin induced nonbacterial prostatitis rat model, and expound the potential mechanisms of LESW therapy. That is novel and meaningful.

As the authors said, duration of capsaicin injection evoked prostatitis is less than 1 week. I wonder whether the authors have considered other prostatitis models, such like autoimmune prostatitis model induced by prostate antigen protein.

During the experiment, energy flux density is the same in all the groups, and so is the frequency. As we know, different energy flux density and frequency could make different results. The Storz shock wave therapy system may have adjustable levers of energy and frequency. Will the authors tell us why the shock number is chosen as the only variable?

Besides, there are some clerical errors in this manuscript, e.g. in Line 24 and Line 258.

Overall, the manuscript presents us a novel and scientifically significant experiment.

Author Response

Response to Reviewer 2 Comments

To my knowledge, there are so many articles about LESW therapy, but few of them explored the mechanisms. In the manuscript, the authors focus on the effect of low energy shock wave (LESW) therapy on the changes of inflammatory molecules and pain reaction using capsaicin induced nonbacterial prostatitis rat model and expound the potential mechanisms of LESW therapy. That is novel and meaningful.

Point 1: As the authors said, duration of capsaicin injection evoked prostatitis is less than 1 week. I wonder whether the authors have considered other prostatitis models, such like autoimmune prostatitis model induced by prostate antigen protein.

Response 1: Currently there is no perfect model to mimic real chronic prostatitis or prostate pain in human. In our group we started using capsaicin prostatitis model for some novel treatments including botulinum toxin and shock wave. We knew this model cannot reflect the long-term condition of prostatitis. Currently we also inject zymosan into prostate to induce a prostatitis model which can last for 4 weeks to further test the effect of shock wave therapy. 

Point 2: During the experiment, energy flux density is the same in all the groups, and so is the frequency. As we know, different energy flux density and frequency could make different results. The Storz shock wave therapy system may have adjustable levers of energy and frequency. Will the authors tell us why the shock number is chosen as the only variable?

Response 2: As we know shock wave therapy has dose dependent effect. According to previous studies, the shock intensity ranging from 0.10-0.13 mJ/mm2 and shock number ranging from 200-300 impulses were the optimal parameters for ESWT to treat cells in vitro (Zhang, X. et al. J Surg Res 186(1) pp484-92). In our pilot studies we used several combination of energy flux densities, range from 0.09-0.2 mJ/mm2.We found 0.2 mJ will sometimes induce some tissue damage in bladder and prostate, and the 0.09 mJ has less therapeutic effect. Therefore, we choose 0.12 mJ/mm2 for our experiment. We choose 120 shocks per minutes as this frequency was commonly used in the previous studies for cystitis and other inflammation conditions. This is the reason why we choose shock number as the only variable.

Point 3: There are some clerical errors in this manuscript, e.g. in Line 24 and Line 258.

Response 3: We have corrected the errors in Line 24 and add citation 13 in Line 258.

Round 2

Reviewer 1 Report

This is well-revised.

Thank you for your efforts.